# Automated Data Acquisition System Using a Neural Network for Prediction Response in a Mode-Locked Fiber Laser

Jose Ramon Martinez-Angulo [1,2], Eduardo Perez-Careta [1,*], Juan Carlos Hernandez-Garcia [1,3], Sandra Marquez-Figueroa [1], Jose Hugo Barron Zambrano [2], Daniel Jauregui-Vazquez [1], Jose David Filoteo-Razo [1], Jesus Pablo Lauterio-Cruz [4], Olivier Pottiez [5], Julian Moises Estudillo-Ayala [1] and Roberto Rojas-Laguna [1]

1   Department of Electronics, DICIS, University of Guanajuato, Carr. Salamanca-Valle de Santiago km 3.5 + 1.8, Salamanca 36885, Gto., Mexico; jr.martinezangulo@ugto.mx (J.R.M.-A.); jchernandez@ugto.mx (J.C.H.-G.); smarkezf@gmail.com (S.M.-F.); jaureguid@ugto.mx (D.J.-V.); jd.filoteorazo@ugto.mx (J.D.F.-R.); julian@ugto.mx (J.M.E.-A.); rlaguna@ugto.mx (R.R.-L.)
2   Faculty of Engineering, Autonomous University of Tamaulipas (UAT), Matamoros SN, Zona Centro, Cd. Victoria 87000, Tamaulipas, Mexico; hbarron@docentes.uat.edu.mx
3   National Council of Science and Technology (CONACYT), Av. Insurgentes Sur 1582, Col. Credito Constructor, CDMx 03940, Mexico
4   Department of Physics Engineering, DCI, University of Guanajuato, Loma del Bosque 103, Lomas del Campestre, Leon 37150, Gto., Mexico; jplauterio@ugto.mx
5   Department of Photonics, Research Center in Optics (CIO), Loma del Bosque 115, Lomas del Campestre, Leon 37150, Gto., Mexico; pottiez@cio.mx
*   Correspondence: perez.e@ugto.mx; Tel.: +52-8341483962

**Abstract:** In this paper, we proposed a system to integrate optical and electronic instrumentation devices to predict a mode-locking fiber laser response, using a remote data acquisition with processing through an artificial neural network (ANN). The system is made up of an optical spectrum analyzer (OSA), oscilloscope (OSC), polarimeter (PAX), and the data acquisition automation through transmission control protocol/internet protocol (TCP/IP). A graphic user interface (GUI) was developed for automated data acquisition with the purpose to study the operational characteristics and stability at the passively mode-locked fiber laser (figure-eight laser, F8L) output. Moreover, the evolution of the polarization state and the behavior of the pulses are analyzed when polarization is changed by proper control plate adjustments. The data is processed using deep learning techniques, which provide the characteristics of the pulse at the output. Therefore, the parameter classification-identification is in accordance with the input polarization tilt used for the laser optimization.

**Keywords:** fiber optics; laser; automation; remote control; neural network

## 1. Introduction

Over the years, automation and data acquisition systems have received special interest due to their potential to solve tasks efficiently in many areas such as medicine [1], manufacture industries [2], video processing [3], agriculture [4], imaging [5], meteorological systems [6], remote laboratory [7], and others. Besides, in some research tasks it is necessary to acquire a considerable number of samples in manual mode by using electronic devices; during this process, the measurements can be affected by errors and information loss for the analysis. The commercial devices for measurements in laboratories are instruments which work directly via recessed user interface and the unique combination of

such devices involves extraordinary efforts to obtain simultaneously data analysis. In many cases, the development of software application in the monitoring and controlling can help reducing time and obtaining samples with good accuracy; here, the use of a graphical user interface (GUI) is a reliable alternative in remote systems [8–10].

According to the literature, in many areas automation represents a tool which provides more efficiency in any control process through the integration of multiple devices [11,12]. On the other hand, it is well-known that the use of software allows the communication through internet media, remote platforms, and interfaces [13,14]. As a consequence, software like LabVIEW is commonly used for creating virtual environments; in addition, this software allows on-line data acquisition or off-line mass data storage through a USB-interface and its subsequent visualization in workstations [15], or in some cases it allows observation and analysis of the experimental processes [16].

With the additional advantage of having a remote laboratory for the simultaneous acquisition and subsequent processing of data, there is also the possibility that such data can be quickly classified or processed using by traditional programming. On the other hand, some advanced algorithms offer the possibility to interpret, self-assess, and learn from the data, such as numerical variables, sound, or medical images [17], which is called machine learning (ML). Categories of ML include supervised learning [18], unsupervised learning, and semi-supervised learning [19]. These algorithms have been widely used for signal processing in applications such as medicine in pattern recognition, analysis [20], and computer vision [21].

In optics research, a large amount of data is handled in the experimental analyses, and only some applications in spectroscopy using genetic algorithms (GAs) for data validation have been proposed [22–24]; in these works, the implementation of a genetic algorithm to intelligently locate the optimal parameters for mode-locking in a single-pulse figure-eight laser (F8L) is experimentally validated taking into consideration wavelength ($\lambda$), frequency (f), time duration (t), and delay ($\tau$). The combining ANN and the dispersive Fourier transform are mentioned in [25] where the authors can predict the pulse duration in a F8L reached until 95% success. Besides, the evolutionary algorithm (EA) is mentioned in [26], where the authors employing this application for control over the high-harmonic spectrum and to shape the laser pulse. Recently in [27], a deep neural network (DNNs) for control and the optimization of the multiple parameters of a deep ultraviolet mode-locked laser is used. In the other hand, in [28], a DNN used as a feature extractor for wide-angle diffraction images of helium nanodroplets is presented; here, the authors use intense short-wavelength pulses generating enable diffractive imaging of individual nanosized objects.

In another work [29], a numerical simulation in combination with GA is studied for a Ti:Sapphire laser system; here, the pulsing is optimized by four parameters: laser, wavelength, peak power, and time duration, and the optimization was carried out using pre-defined functions by the GA in MATLAB. Also, in [30] it is mentioned how the GAs have been used to optimize the parameters (wavelength, temporal width, and peak power to generate supercontinuum spectra.

In our study, the integration and automation of optical instrumentation through LabVIEW for remote data acquisition is carried out, and a neural network system with supervised learning for the presentation of results is implemented. The data samples are extracted from a F8L output when the polarization plates located in the nonlinear optical loop mirror (NOLM) [31], that acts as a saturable absorber, are affected by varying the polarization controller between 0 and 180°.

## 2. Materials and Methods

### 2.1. Laser Setup and Data Handling

The experimental scheme used for generating the data consists of a F8L which operates in continuous wave (CW) and mode-locking pulsed regimes; the scheme is shown in Figure 1. The F8L is pumped by a laser diode with a spectrum centered at 979 nm and an input power of 3.7 W. This laser is composed by a ring cavity and a NOLM. In the former, the pump and the signal feedback

are connected by an optical combiner (980/1550 nm), the loop includes an active medium (3.5 m of a Er3+/Yb3+ double-clad fiber (EYDCF)); core diameter of 12 μm and inner cladding diameter (flat to flat) of 130 μm, where the core exhibits an absorption at 1530 nm. A polarization controller with two polarization plates (a quarter-wave retarder (QWR, λ/4) [32] and half-wave retarder (HWR, λ/2)) is used to adjust the polarization state in the cavity. Besides, a polarization-dependent isolator (PDI) is used to fix the polarization state and the unidirectionality of light control. After the first coupler (polarization monitoring no. 1 (99:1)), the HWR is set for polarization control. A symmetric (50:50) coupler is used to build the NOLM (which is a power-symmetric, polarization-imbalanced scheme, as the saturable absorber), which promotes the mode-locking and generation of optical pulses. The NOLM includes 5-m-long fiber twisted at a rate of 5 turns per meter to decrease the residual birefringence and maintain circular birefringence. This fiber is connected with a QWR (PC1) to control polarization inside the NOLM, which can control the nonlinear switching characteristic. The QWR plate is set at 45° in order to transform the linear input polarization light into circular polarization; at the NOLM output (transmission), a third coupler is connected (polarization monitoring reference no. 2 (95:5)), to estimate the transmission; right after the 95% coupler output port, 200-m-long single-mode fiber (SMF-28) is connected, which constitute most of the resonant cavity of the laser. When the cavity is closed an optical fiber coupler (80:20), here, the port (80%) is feedback into the system and the port (20%) is subsequently connected to a coupler (50:50) that is used as final output monitoring (no. 3 and no. 4), these ports are used for the connection of the optical instrumentation. The laser operates with a repetition rate of 940.1 kH (which means a period of 1.0638 μs) in a total length cavity of ~209.96 m. It is important to mention that the average output power of the laser in pulsed operation is ~10 m when it is pumped by 3.7 W. This means that the laser efficiency is less than 1%.

The F8L outputs are connected to the optical instruments in the laboratory, so that, the system is connected to a data bus through TCP/IP protocols to perform the data acquisition with each of the optical instruments: polarimeter (TXP, Thorlabs PAX57101R3-T); optical spectrum analyzer (OSA, Anritsu MS9740A); oscilloscope (OSC, Keysight MSO6004A). They are synchronized by the National Instruments Software Measurement and Automation Explorer (NI-MAX) in a computer; through automation, the data is acquired simultaneously from each of the instruments and processed by the GUI developed in LabVIEW for automatic storage and processing of the information. Subsequently, all the data obtained by the integration are saved and processed (using the MATLAB software) to display the results and use the neural network for classifications of data.

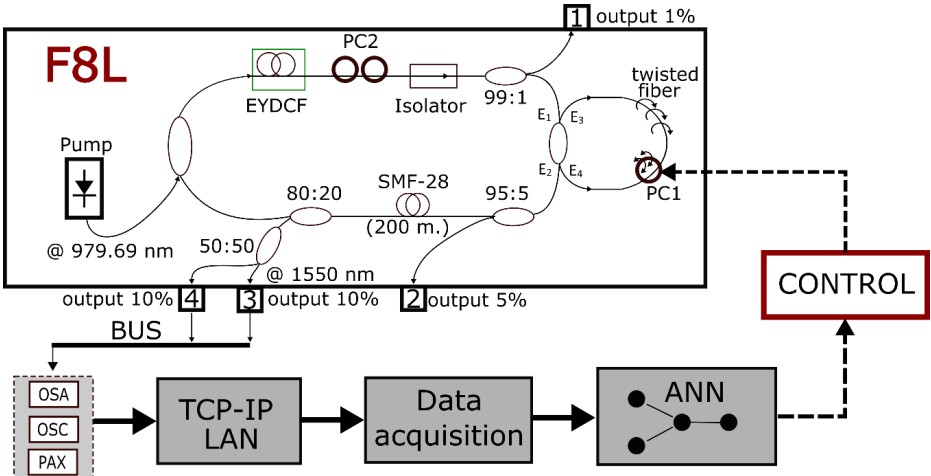

**Figure 1.** F8L experimental scheme for automated data acquisition remotely.

The schematic diagram of communication and data acquisition is depicted in Figure 2. It is composed by a router (Linksys Wireless-B 2.4 GHz) as a central communication system, which allows connecting the optical instruments through the TCP/IP LAN protocol, defining an IP address for each

instrument; this connection synchronizes the optical instruments for the acquisition of information from the laser output (temporal pulses, spectra, and polarization state data). All the data are saved directly into a local drive (computer) through the interface created in LabVIEW; this allows manipulating the acquired samples (with user-defined intervals since ~10 ms ). Moreover, the configuration and characteristics of instruments can be easily changed remotely for simultaneous data acquisition.

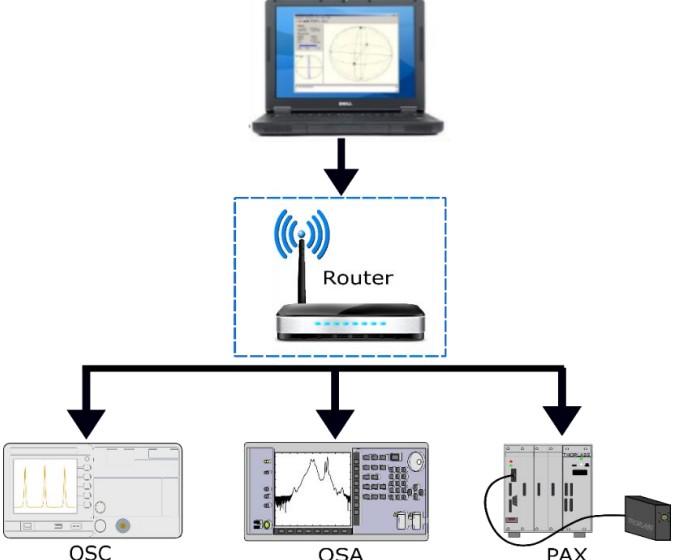

**Figure 2.** Schematic diagram base for remote communication via LAN with the optical instrumentation.

### 2.2. Artificial Neural Networks (ANN)

These systems can learn, perform tasks, and consider examples, including some with not previously defined or applied specific rules. The algorithm makes its own decisions using examples or patrons with auto-learning [33]. With some defined parameters, an ANN can learn fast and can control them, and in addition, they can be used for systems optimization [34]. Besides, it is important to mention that ANNs are usually employed in some areas of engineering to recognize certain phenomena and predict diseases [35,36].

In our study, the engineering steps are depicted in Figure 3a, and the data processing schematic implemented with an ANN is shown in Figure 3b. Here, the variables used to determine the fiber laser response are presented. To ensure the optimal performance of a laser, it is necessary to consider the parameters to be controlled as well as the optical devices used for its characterization and measurement. In our case, a passive mode locking (PML) F8L generates pulses based on the NOLM switching [37]. Therefore, the use of algorithms helps to obtain faster and more accurate results to control key devices in the laser configuration [23].

The information of the critical variables for the locking of modes used for the feature classifications of the laser output was based on the following: the pulse temporal shape at the laser output was measured using an InGAS photodetector (Thorlabs DET01CFC) connected to the OSC instrument, capturing a full period (a round trip time) of the resonant cavity, which include one pulse. The acquisition of the spectrum was performed by an OSA and finally, the polarization is monitored by a PAX. PC1 is performed from 0 to 180 degrees with 10-degree intervals, making simultaneous acquisitions of the generated spectra, temporal pulse, and polarization states at each point of the variable plate. The Stokes polarization parameters vectors are defined into the algorithm as S1, S2, S3, and the angle of the polarization plate is defined as target 'Ang'. Therefore, output parameters were defined as target 1, target 2, and target 3. These parameters represented the polarization type, spectral width, and pulse width, respectively. The confusion matrix is the same for each of the mentioned cases and the final

parameters can change when required for the user. In this case, we focus on the confusion matrix with polarization data for control with the plate QWR located in the F8L.

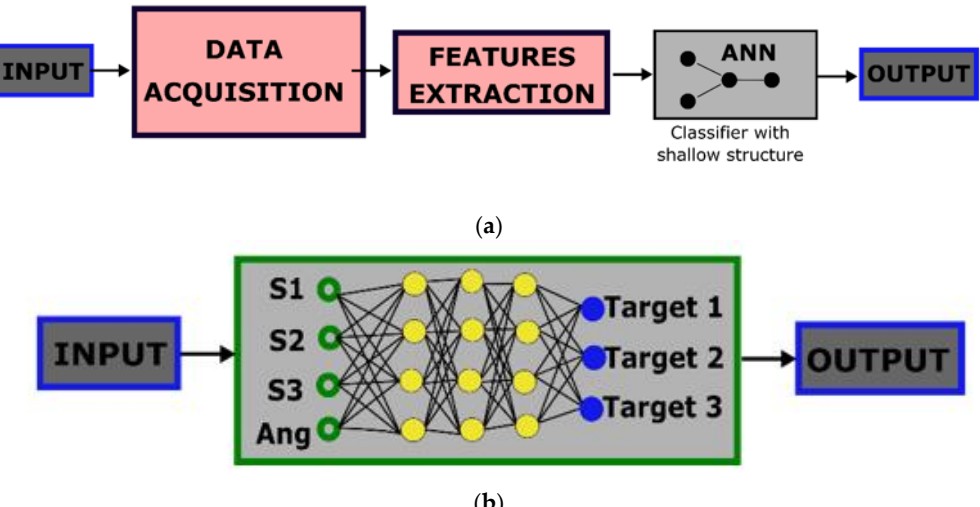

(a)

(b)

**Figure 3.** Schematic diagrams: (**a**) general processing data system extraction + selection, (**b**) neural network features learning + classifier.

In this work, cases used for the analysis of data by the ANN and the method used to classify and predict information are based on the Choen's Kappa coefficient, mentioned in [38]. Therefore, base equations and types of indices are given by

$$P = \frac{TP}{TP + FP} \tag{1}$$

$$rCall = \frac{TP}{TP + FN} \tag{2}$$

$$F1 = 2\frac{P * rCall}{P + rCall} \tag{3}$$

$$Ac = \frac{TP + TN}{TP + TN + FP + FN} \tag{4}$$

$$TPrate = \frac{FN}{FP + FN} \tag{5}$$

$$FPrate = \frac{FN}{FP + FN} \tag{6}$$

$$TNrate = \frac{FN}{FP + FN} \tag{7}$$

$$FNrate = \frac{FN}{FP + FN} \tag{8}$$

where *TP* the true positive value, *FP* is the false positive, *FN* is the false negative, and *TN* is the true negative value. Considering these output parameters, it is possible to evaluate the precision (*P*), the recall (*rCall*), the F1-score (*F1*), the accuracy (*Ac*), the true positive rate (*TPrate*), the false positive rate (*FPrate*), the true negative rate (*TNrate*), and false negative rate (*FNrate*). Equations (1)–(8) offer an overview and great advantages for modeling these data types, as mentioned in [39,40].

We use two types of data classes in this algorithm, where the false positives predict when the event happens, but it does not actually happen, this will be a 'type 0' error. A false negative which predicts that something will not happen, but it actually happens, is called 'type 1'; any other type is an elliptical polarization event. The algorithm uses these two classes to know the type of final

polarization type accord the input declared (called S1, S2, S3, or stokes values [41]) besides, the value of the 'ang' provides information about the variation at the output and subsequently to stokes parameters values, as a result the changes in polarization modes can be estimated. The parameter S0 describes the total intensity of the optical field and was declared with the value of one in the algorithm, due to the total sum of components is equal to the unit. The PAX get the stokes parameters called (stokes 1, 2, and 3), and using these parameters, we can calculate the polarization state into the fiber laser cavity.

## 3. Results

The implemented system interprets the acquired results remotely through the ethernet network (TCP/IP). The increase in the angle for data acquisition consist in varying the retarder plate PC1 (located within the NOLM); for proper configuration of PC1, we generate harmonics mode-locking pulses [42] with repetition frequencies of 940.1 kHz up to 1.8 MHz. For simultaneous acquisition with the instrumentation, we use a 50:50 coupler at the output of the 80:20 coupler. The sampling allowed observing the evolution of the polarization, simultaneously detecting the components of the temporal and spectral waveform and, at the same time, saving the information in the remote storage unit for later processing for the classification and identification of the parameters, as well as the optimization of the laser using the ANN.

The OSC screen display is shown in Figure 4a. This interface developed through virtual instrumentation (VI) in LabVIEW for the remote acquisition of data using the TCP-IP LAN protocol [43]. The interface allows acquiring the samples remotely, which can be visualized and processed by the virtual environment (OSC) on a computer. Remote communication for data acquisition is possible through a National Instruments (NI) controller and the Virtual Instrument Software Architecture (VISA) [44]. The I/O standard for the instrument was carried out with the TCP/IP addresses using a router for integration and administration, as mentioned before.

The virtual environment allows setting the configuration and specifications of the instrument remotely and manipulating the characteristics of the signal measurement. The number of samples/seconds are defined in the GUI and can be monitored and personalized in real-time. The real-time acquisition of the data provided by the OSA is shown in Figure 4b. Here, the spectral data environment remotely implemented via LabVIEW is presented. The full Figure 4 shows the general block diagram process of environment VI from acquisition configuration to data saving for interface environment remotely for the instruments.

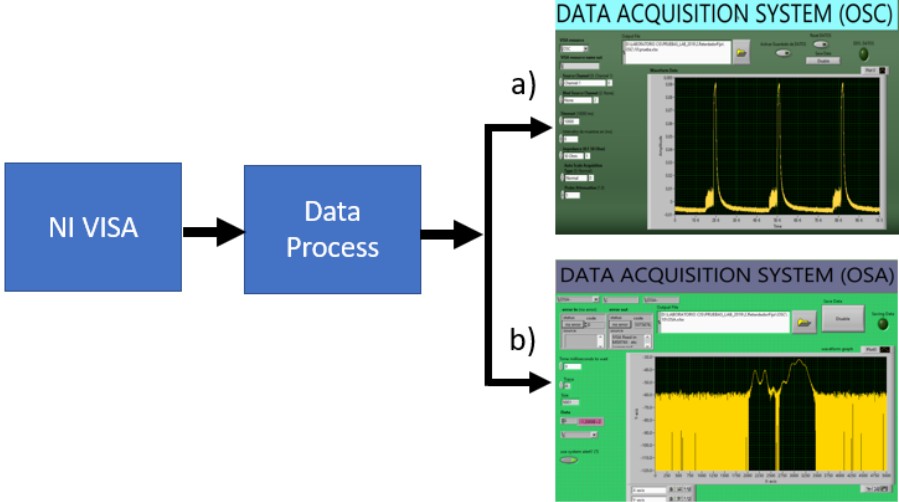

**Figure 4.** Graphical interface data environment remotely system: temporal (**a**) OSC and spectral (**b**) OSA.

The polarization state and the data acquired simultaneously by the instrument for 120° plate adjustment is shown in Figure 5a. The acquired data are depicted in Figure 5b,c (temporal) and Figure 5d,e (spectral), respectively. Samples acquired through the remote network are plotted through MATLAB. For simultaneous acquisition tests, PC2 are still fixed, and the variation was performed only in PC1. The data showed presents stability with a predominant elliptical right polarization, a pulse width at the full-width half maximum (FWHM) [45] of 39.76 ns and a spectrum centered at ~1545 nm. Through the data obtained, an evolution in the temporal and spectral components was observed: an increase in the pulse width with elliptical polarization and an increase in the amplitude when the elliptical parameter is close to the linear state polarization. For this, 100 samples were obtained at 16 ms intervals at the laser output with the elliptical predominant polarization (right).

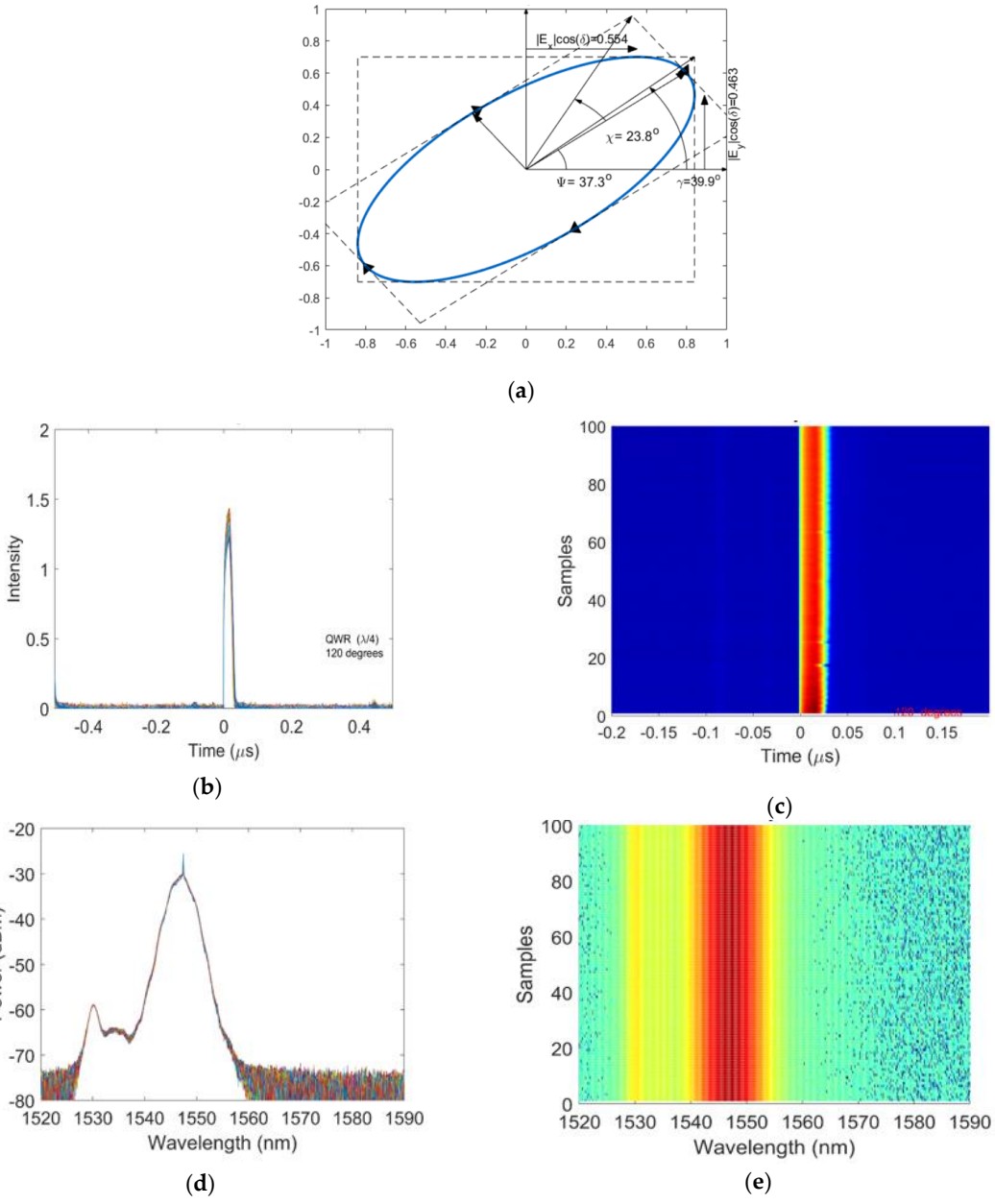

**Figure 5.** Acquired data simultaneously in real-time for the neural network: (**a**) polarization (right elliptical polarization at 120°), (**b**) temporal pulse profiles, (**c**) temporal evolution (100 samples), (**d**) spectra, and (**e**) spectral evolution (100 samples).

Data analysis requires visualization and processing of acquired data, as well as the integration of instrumentation through system automation. The data are processed using the ANN through a supervised technique to show features of the signal at the laser output and its classification. The background noise in Figure 5b–e is due to experimental measurements through the photodetector used to detect the optical signal for the OSC (2.5 GHz in real-time, Keysight MSOX6004A, Keysight Technologies, Santa Rosa, CA, USA).

By means of the ANN, the results are exhibited as follows: the inputs represented the features and the outputs represented the targets (Figure 3b). Firstly, the ANN was created using the MATLAB Neural Network Tool. To recognize certain patterns and for each input feature, a neuron is required in the ANN. Thus, the neural network configuration has four neurons at the input. Then, there is an inner layer of 20 neurons. To construct the ANN with the division of data Diverand technique, it was used the training type Levenberg–Marquardt (LM) algorithm and the MSE and calculations type MEX [46]. Commonly, this technique has been used in prediction systems to reduce the data analysis time [47,48] and to help with the optimization systems [34]. Besides, the LM algorithm has been used in several works to provide a numerical solution minimizing a nonlinear function [34,49,50]. In Figure 6, the cases taken by confusion matrix in two classes are shown; the matrix frames where the error is relatively small, mainly in training and tests, obtaining a value of practically zero in the validation. The first 24% of the data belong to class zero, which represents the elliptical polarization (there are no false negatives or false positives). From the second class, we have 19 data accounting for 76% of the total data representing the linear polarization. The data classes into the matrix represent 100% of the data. There is a 100% on validation for all cases because there are very linear values and no outliers that can affect the behavior of the distribution of data in the system. Here, we show the false positives or type 0 located at the top right, and at the left part, we will have the false negative for the type 1.

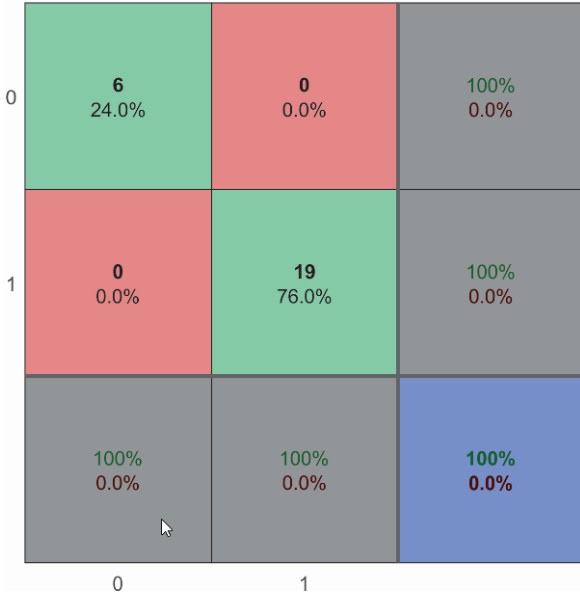

**Figure 6.** Confusion matrix, where 0 is circular and 1 is lineal polarization.

The confusion matrix is used to evaluate the quality of the output of an ANN. Here, the results are as good as they could be, given the following observed values: positive predictive values, PPV = 100%, negative predictive values, NPV = 100%, relative risk, RR = 100%, and screening round, SR = 100%. The algorithms do not identify outliers, so these elements are not representative of the study since they do not affect the behavior of the data. Although excellent results were obtained with the algorithms, the study is not exempt from these values. It is important to mention that continuous performance will be conditioned by the default values.

The data obtained at the laser output mainly depends on the configuration and use of optical devices connected in the cavity and arrangement for experimentation. The length, the pumping power, the type of fiber used, each of them has an effect and can be yielding different characteristics at the output, but these variables can be eliminated and replaced according to the case.

The performance of the ANN can be visualized in Figure 7, where the final weights after training the network are observed. The performance of the ANN shows training (100%), accuracy (100%), and test (100%), which can be viewed in the figure. The neural network training performed three iterations, showing high validation accuracy and high efficiency. Therefore, it must be concluded that the network can follow the pattern of the data as well as memorize the training data; it can be said that overtraining has occurred due to results. In this regard, it is also important to mention that the model is useful for the initial purpose of predicting behavior in the unobserved parts, given that the values are ideally distributed, as the measurement has no noise or bias to be eliminated.

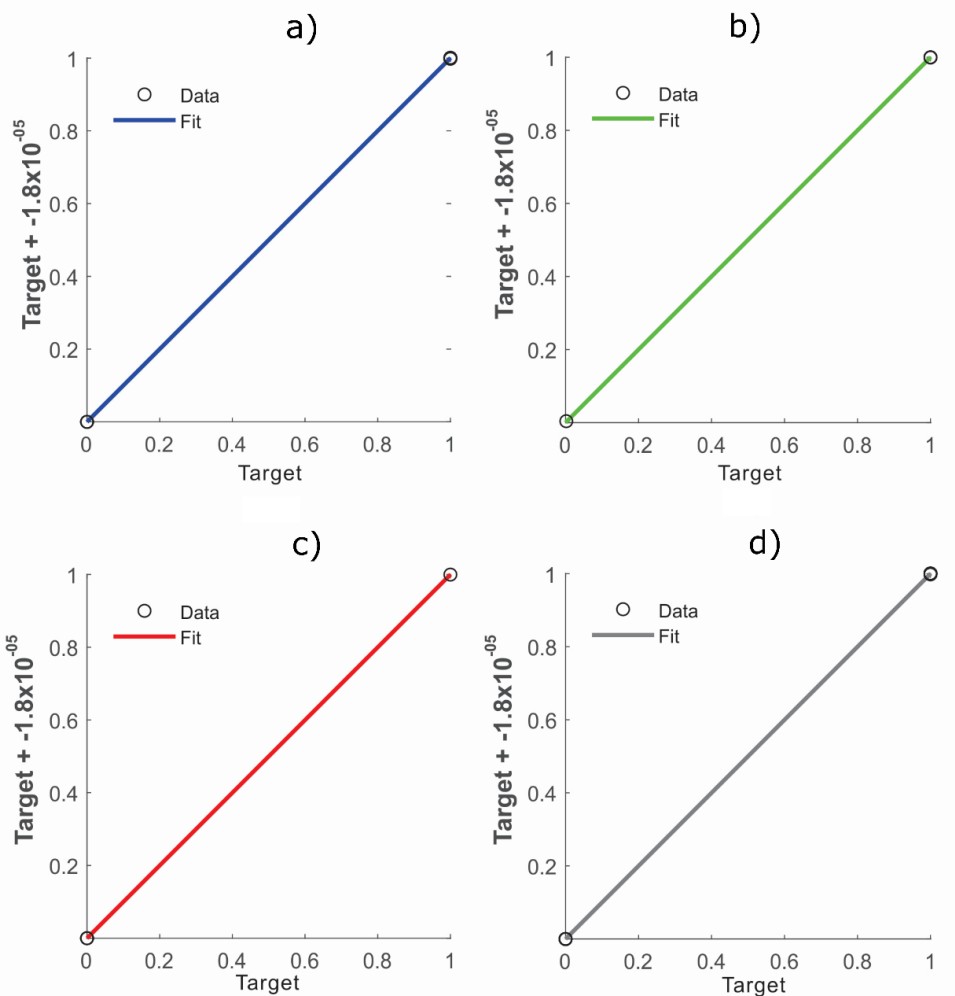

**Figure 7.** Neural network training system using 80 input data (ns): (**a**) training, (**b**) validation, (**c**) test, and (**d**) all parameters.

The results of the errors between the interactions are shown in Figure 8, where the best test and validation optimal value is three given by the algorithm was observed. It can be seen that the process succeeds in reducing the root-mean-square (RMS) error until the model has been adjusted with the data. It can also be observed that if the number of interactions is increased, the ANN decreases its performance. The RMS error given for the validation and test was of $3.0241 \times 10^{-8}$ [51]. As can be appreciated in Figure 7, the acquired data present minimal variation in its response, as a result, this algorithm represents a very useful tool that facilitates training. Moreover, according to Figure 8,

good precision and reduction of the RMS can be expected. It can be observed that the data is correlated and perfectly classified by the ANN implemented, consequently, the algorithm used can be validated. In this case, the data has a deterministic behavior; if the case were different, another classifier would have to be used.

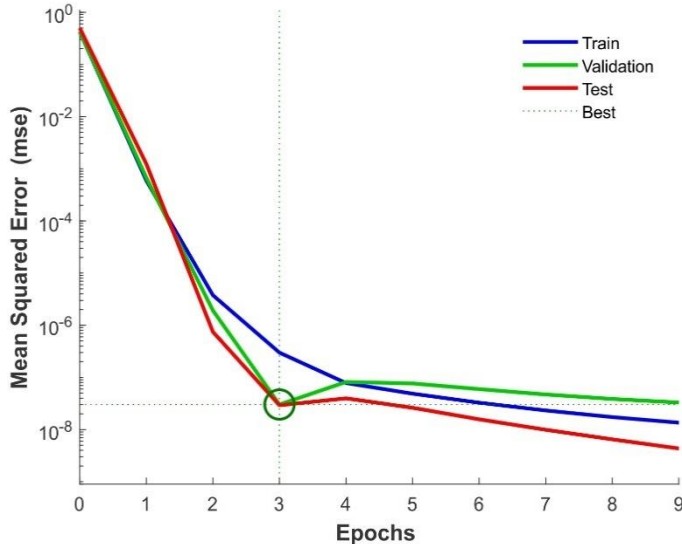

**Figure 8.** Best validation performance using the ANN for processed data reliability.

The system of the neural network allows classifying and finding the optimal fit values on the polarization plate and thus, determining the most stable areas for the generation of pulses at the F8L output.

## 4. Conclusions

In summary, a remote monitoring and control system for data acquisition through the integration of optical instrumentation was demonstrated. Furthermore, this system was used in the acquisition of simultaneous data; processing was related using the implementation of an ANN to determine the characteristics of the signal at the laser output. It was discovered that, by considering the laser input parameters such as polarization and power, it is possible to predict quickly the adjustment parameters of the retarder plates located in the saturable absorber section (NOLM) for generating the mode-locking regimen in more stable ways. Also, by using the ANN, it is possible to do feature classification and prediction at output laser signal, as well as the parameters for the supercontinuum spectra and nonlinear effects generations.

This remote interface is useful to obtain control of and coordinate the data acquisition administration and also, to report of results in a process by required high precision and long working hours of monitoring in the laboratory. The neural algorithm described the laser outputs both in CW and in mode-locking regime. This study shows the characteristics of the light through a classification of patterns from the laser, and at the same time, use these same parameters to control and optimize the light emission. Data analysis with an ANN provides a quick tool for the identification of pulse shape and behavior, in order to obtain and manipulate important characteristics to be used in supercontinuum generation or in the study of complex dynamics.

**Author Contributions:** Conceptualization, J.C.H.-G., S.M.-F., and J.H.B.Z.; Data curation, J.R.M.-A.,E.P.-C., and S.M.-F.; Funding acquisition, J.C.H.-G., J.M.E.-A., and R.R.-L.; Methodology, J.R.M.-A., J.P.L.-C., and J.H.B.Z.; Project administration, J.C.H.-G., J.D.F.-R., and O.P.; Software, J.R.M.-A., E.P.-C., and J.D.F.-R.; Supervision, J.C.H.-G., R.R.-L., and D.J.-V; Validation, J.M.E.-A., O.P., R.R.-L., and D.J.-V.; Writing—review and editing, J.R.M.-A., J.P.L.-C., and S.M.-F.; Writing—original draft, J.R.M.-A.; Visualization, J.R.M.-A. All authors have read and agreed to the published version of the manuscript.

**Funding:** This work was funded by the CONACYT project Catedras CONACyT project no. 3155, CONACyT 'Ciencia Básica' project no. 257691, CONACYT with the projects no. CB-2016-01-286916 and no. A1-S-33363/CB2018 also DAIP no. CIIC 230/2020 and CIIC-238/2020 by Universidad de Guanajuato.

**Acknowledgments:** The Universidad Autónoma de Tamaulipas with fellowship no. 511-6/17-6886 and Catedras CONACyT project no. 3155, CONACyT 'Ciencia Básica' projects no. 257691, also we acknowledge support from the J.M. Estudillo-Ayala with project CONACYT projects no. CB-2016-01-286916; no. A1-S-33363/CB2018 also DAIP no. CIIC 230/2020 and CIIC-238/2020 by Universidad de Guanajuato.

**Conflicts of Interest:** The authors declare no conflict of interest.

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
