# Peer review of "Automated Data Acquisition System Using a Neural Network for Prediction Response in a Mode-Locked Fiber Laser"

_electronics, doi:10.3390/electronics9081181_

Round 1

Reviewer 1 Report

The manuscript “Automated data acquisition system using a neuronal network for prediction response in a mode locked fiber laser” by J.R. Martinez-Angulo et al. deals with the implementation of an ANN for mode-locking of a fiber laser. This topic is interesting for publication in Electronics. However, several aspects of the manuscript need to be improved.

1. The introduction misses some important papers in the field of lasers and ANN. E.g. "Machine learning-based pulse characterization in figure-eight mode-locked lasers," Opt. Lett. 44, 3410-3413 (2019) , A Deep Ultraviolet Mode-locked Laser Based on a Neural Network. Sci Rep 10, 116 (2020). https://doi.org/10.1038/s41598-019-56845-6

The authors should add a paragraph to the introduction showing the current field of research in lasers and ANN.

2. There are more prominent examples about the application of generic algorithms. E.g. everything related to femtochemistry, pulse shaping and Ahmed Zewail. E.g. Extremely Nonlinear Optics Using Shaped Pulses Spectrally Broadened in an Argon- or Sulfur Hexafluoride-Filled Hollow-Core Fiber. Appl. Sci. 2015, 5, 1310-1322.

3. A good example for image processing applications would be “Deep neural networks for classifying complex features in diffraction images” Phys. Rev. E 99, 063309.

4. Equations 1 to 8 need an explanation. What are the variables?

5. Figure 4a and 5a are not necessary. Figure 4b and 5b should be figure 4a and b.

6. The block diagram in Figure 6 does not contain any important information, since everything is in the subVIs. I do not understand why the authors emphasize the use of Labview that much. While the implementation of a GUI in Labview is simple and most hardware can be accessed using NI VISA, this is no novelty. In principle everything could be implemented in Python or any other programming language.

8. What does Figure 9 show? All plots show two datapoints (0,0) and (1,1) and a straight line.

9. Also Figure 10 needs more explanation.

10. The authors should review their reference list. Many papers are conference proceedings that are more than 10 years old. There overview / review articles should be used instead. The total number of almost 60 references seems to be too much.

Reviewer 2 Report

In the article « Automated data acquisition system using a neural network for prediction response in a mode-locked fiber laser », J.R. Martinez-Angulo et al have described the use of a neural network for optimization of a figure-8 fiber laser. Though the main idea of the paper seems to be relevant due to the nonlinear behaviour of such system, there is an overall lack of information at the end of the article while previous parts in the paper are useless. The English usage is not perfect. The paper cannot be accepted and should address the following points for publication.

In more details:

  1. Needed scientific details

  • ANN construction, how did they decide to use only 20 neurons and why?
  • Line 233, Training type Levenberg-Marquardt, lack of information on this algorithm, comparison with other ANN methods, what are the advantages and disadvantages of this method and the reason why it has been chosen.
  • Line 235, Confusion matrix, at this point, there is no information about the whole process. They do not explain if there is a preprocessing in the input layer (S1, S2, S3 and Ang), training and testing is loosely described. Moreover, length of the cavity, pumping power and the type of fiber used are critical parameters in operation of the laser and these cannot be eliminated as explained in line 256. These parameters play a very important role in finding the mode-lock operation of the laser.
  • Line 287-290, the authors talk about the power of the laser but this one was never mentioned before in the setup. It is clear that considering it like polarization would perhaps make it possible to predict the setting of parameters for the generation of the mode-lock operation of the system. However, at each power value, a monitoring of the system would have to be made: pulse shape, pulse length and polarization. Then applying an ANN method would not be so simple with the one presented in this article, because the number of variables would be larger and therefore the inner layer would be made up of more neurons. But, it would be something much better and more general in the classification and prediction of the response for the mode-lock fiber laser.

  1. Shortening

Page 6 and page 7 have to be shortened. In particular the figures (4, 5, 6) are completely useless for the general understanding of the paper. In a general way the authors take too much time for description of the acquisition system. 

  • Problem of English usage and technical problems

Here is a non exhaustive list of problems met in the document

  • Line 41 (English), “be affected be errors”
  • Line 76 (technical point), “0 and 190°”. Why this angle has been chosen ? Are 180° not enough for describing a complete cycle?
  • Line 100,“321 kHz repetition rate and with a period of 1.1152us”, period and repetition rate are not consistent if period means the time between two subsequent pulses.
  • Line 104, problem of “()”
  • Line 130, “can controlled”
  • Line 169, “the acquires results”
  • Figure 4, the optical pulses don’t look clean. There is an oscillating part stuck to the pulse which doesn’t look to the typical behavior of a photodiode.
  • Figure 7c, “temporal evolution (100 samples) while only 50 are shown
  • Line 224, laser power is 10 mW when pumped at 3.7W pump power. The efficiency is well below 1%, this should be commented as fiber laser generally exhibit much better efficiency.
  • Line 234, “this technique have been”
  • Line 256, “but it is variables”
  • Line 273, “the ANN decreases it is performance”
  • Line 290, “regimen”

Reviewer 3 Report

The authors demonstrate a system for integrating optical and electronic instrumentation devices for prediction of a mode-locking fiber laser response, using a remote data acquisition with processing through an artificial neural network. The presented results are well described and can be replicated. Machine learning is a rapidly devolving field of knowledge and its application to engineering problems, in particular in the laser optimization, is of great interest. Therefore, the work may be interesting for a wide range of readers. I have only minor suggestions.

1) It would be interesting to see calculated spectral phase.

2) The authors repot mode-locked laser with a broad spectrum but with ns duration. It should be clarified in more detail why this regime is preferable here.

3) Line 41, there is a typo: “be affected be…”

Round 2

Reviewer 1 Report

The authors modified the manuscipt according the suggestions of the reviewers. I recommend the manuscript for publication.